# DualResearch: Entropy-Gated Dual-Graph Retrieval for Answer Reconstruction

## Abstract

The deep-research framework orchestrates external tools to perform complex, multi-step scientific reasoning that exceeds the native limits of a single large language model. However, it still suffers from context pollution, weak evidentiary support, and brittle execution paths. To address these issues, we propose **DualResearch**, a retrieval and fusion framework that matches the epistemic structure of tool-intensive reasoning by jointly modeling two complementary graphs: a *breadth semantic graph* that encodes stable background knowledge, and a *depth causal graph* that captures execution provenance. Each graph has a layer-native relevance function, seed-anchored semantic diffusion for breadth, and causal–semantic path matching with reliability weighting for depth. To reconcile their heterogeneity and query-dependent uncertainty, DualResearch converts per-layer path evidence into answer distributions and fuses them in log space via an *entropy-gated* rule with global calibration. The fusion up-weights the more certain channel and amplifies agreement. As a complement to deep-research systems, DualResearch compresses lengthy multi-tool execution logs into a concise reasoning graph, and we show that it can reconstruct answers stably and effectively. On the scientific reasoning benchmarks HLE and GPQA, DualResearch achieves competitive performance. Using log files from the open-source system InternAgent, its accuracy improves by 7.7% on HLE and 6.06% on GPQA.

## 1 Introduction

Large language models (LLMs) have demonstrated remarkable capabilities and provided new paradigms for tackling scientific tasks across various domains (Achiam et al., 2023; Zhang et al., 2025). However, current LLMs still lack explicit chains of evidence, systematic reasoning processes, and structured modes of knowledge organization in scientific reasoning applications (Shojaee et al., 2025). As a result, their responses often fall short in terms of reliable theoretical grounding and logical rigor. Moreover, native models face challenges when integrating long-text information: they struggle with global planning (Li et al., 2024), cross-paragraph alignment (Huang & Chang, 2023), and consistency maintenance (Ahmed & Devanbu, 2023). In other words, complex scientific tasks are often difficult to resolve through a single round of reasoning (Zhang et al., 2025).

To address this, a set of approaches known collectively as *deep-research* has been proposed (Jones, 2025; Hu et al., 2025; Team et al., 2025). These methods integrate LLMs with external information retrieval and tool usage, enabling models to acquire and incorporate external knowledge during reasoning. When necessary, they can decompose complex tasks through multi-agent collaboration, and attach explicit citations of evidence in their outputs, thereby enhancing, to some extent, their ability to solve challenging scientific problems. As illustrated in Figure 1, when answering the question *"Which of these Turing Machines halts after the most number of steps and what is the number of steps?"*, a typical deep-research workflow retrieves the transition tables of the relevant Turing machines, generates corresponding simulator code, and ultimately derives a conclusion.

Nevertheless, these methods still exhibit failure modes. First, noise introduced by semantic retrieval may mislead the simulation logic (Shi et al., 2025). Second, conclusions are often presented without an explicit demonstration of intermediate steps (Prystawski et al., 2023). The root cause is that deep research operates as a tool-intensive paradigm. It requires broad semantic anchoring across concepts, aliases, and cross-literature evidence (Huang et al., 2025).

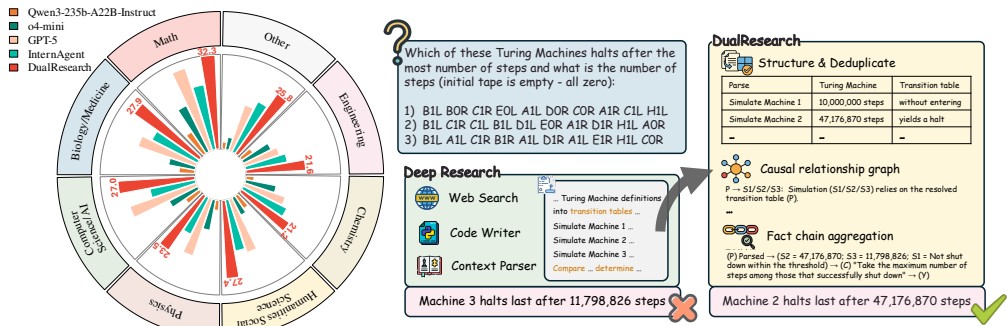

Figure 1: The illustrations for DualResearch. Left: Performance comparison on the HLE benchmark, where DualResearch consistently outperforms strong baselines across diverse scientific domains. Right: A case on the "Turing Machine halting steps" problem, where deep research produces an incorrect conclusion due to noisy retrieval and missing causal constraints, while DualResearch leverages structured process graphs and entropy-gated aggregation to derive the correct answer.

Based on this, we argue that retrieval and aggregation should reflect the epistemological structure of the task. To achieve this, we propose a new framework termed **DualResearch** constructing in parallel a *Breadth Semantic Graph*, which organizes semantic and evidential connections among entities, paragraphs, and tables, and a *Depth Causal Graph*, which encodes causal reasoning through typed actions, outputs, and verifiers. During reasoning, problem-solving advances through two collaborative stages: (1) breadth-oriented neighborhood expansion, where multi-hop semantic propagation around seed terms suppresses drift; (2) depth-oriented causal constraint analysis, where short and reliable execution chains are selected.

To further unify the semantic and procedural knowledge, we introduce an entropy-gated dual-graph fusion. Each side first forms a normalized answer distribution, which we then fuse in log space with entropy weights, giving more weight to the sharper (more certain) distribution. This boosts agreement while avoiding overconfidence under joint uncertainty. The fused output preserves recall from similarity retrieval and produces stitchable chains of evidence for the LLM to leverage. With dual-graph modeling and entropy-gated fusion, this work advances scientific reasoning from *Similarity-based Paragraph Matching* to *Causality-based Verifiable Reasoning*, improving reliability, traceability, and reproducibility.

In summary, our contributions are as follow:

1. To address the trade-off between semantic coverage and causal consistency in complex problem solving, we propose DualResearch. To our knowledge, this is the first framework that jointly models *Breadth Semantic Graph* and *Depth Causal Graph*, which assigns each Graph its own layer-native relevance function: seed-anchored semantic diffusion for breadth, and causal-semantic path matching with reliability weights for depth.

2. We propose an entropy-gated fusion mechanism that transforms evidence from both graphs into answer distributions. The mechanism reconciles two heterogeneous signals, undirected semantic neighborhoods and directed procedural paths, thereby ensuring robustness under channel disagreement and enhancing performance when the two channels align.

3. On graduate-level scientific datasets HLE and GPQA, our method successfully reused the log of the baseline and achieved superior performance. Moreover, when compared with state-of-the-art methods, DualResearch also demonstrated competitive results.

## 2 RELATED WORK

**Retrieval-Augmented Generation (RAG)** enriches LLM prompts with external evidence so that responses are grounded in factual sources (Ram et al., 2023; Fan et al., 2024). Standard RAG retrieves top-$k$ text chunks from a vector index, which is effective for short-hop fact lookup but fragments documents and offers no representation of the reasoning process (Gao et al., 2022; 2023; Chan et al., 2024; Yu et al., 2024). Recent graph-based RAG begins to link entities or claims (Edge et al., 2024), yet most methods remain text-oriented and still lack an explicit channel for procedural

information, limiting reproducibility and multi-step reasoning. Beyond RAG, a complementary literature study examines how graphs interface with LLMs and agents. Three strands are prominent: (i) using graph neural network (GNNs) (Han et al., 2022) to produce topology-aware tokens for LLMs, *e.g.*, GraphGPT (Tang et al., 2024) and LLaGA (Chen et al., 2024); (ii) using LLMs to enrich graph content and provide supervision for downstream tasks, e.g., GALM (Xie et al., 2023) and OFA (Xie et al., 2023; Liu et al., 2024); and (iii) building agents that directly operate on graphs to align GNN and LLM representations through interaction (Li et al., 2023; Brannon et al., 2023).

In contrast, standard RAG assumes a fixed corpus and falters when answers require dynamic tool use. Our approach treats graphs as execution objects: a breadth channel anchors terms across documents, while a depth channel retrieves short, auditable procedure chains produced during tool interaction. The result is not just content-grounded answers but tool-grounded and reproducible reasoning, addressing cases where chunk-based RAG is brittle.

**Deep-Research** motivates a series of systems for scientific problem solving. OpenAI (2025c) and DeepMind (2024) combine retrieval with reasoning to generate evidence-grounded reports from heterogeneous sources. Building on multi-agent coordination, OWL (Hu et al., 2025) employs a hierarchical architecture, while InternAgent (Team et al., 2025) extends this paradigm with closed-loop workflows for iterative hypothesis generation and experimentation. WebThinker (Li et al., 2025) emphasizes dynamic web-based reasoning, integrating preference optimization for long-horizon inference. In contrast, single-agent approaches such as SFR-DR (Nguyen et al., 2025) train LLMs to select actions via reinforcement learning, whereas X-Masters (Chai et al., 2025) advances ensemble reasoning through a multi-channel strategy. Together, these systems highlight the diversity of agent designs but also reveal open challenges in controlling solution space and ensuring reproducibility.

Despite these advances, large-scale retrieval and tool use expand the solution space and amplify uncertainty. Our approach instead leverages solving logs to distill declarative facts and procedural steps, thereby narrowing the solution space while improving transparency and reproducibility.

# 3 METHOD: BREADTH & DEPTH GRAPHS WITH LAYER-NATIVE RETRIEVAL DISTANCES

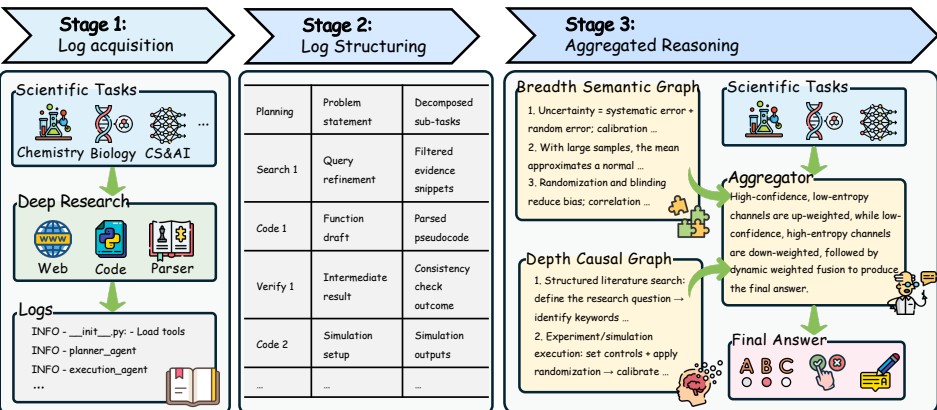

Figure 2: Workflow of the DualResearch. Stage 1: scientific tasks are executed with Deep Research tools to produce raw logs. Stage 2: logs are structured into stepwise traces with intermediate artifacts. Stage 3: evidence is organized into a Breadth Semantic Graph and a Depth Causal Graph, whose outputs are fused by an entropy-gated aggregator to yield the final answer.

**Background.** Tool-intensive questions usually need multi-hop reasoning before obtaining the answer. The model must ground terms and assumptions in background sources and then compose results across tools *(search→parse→compute→verify)*. Text-only graphs capture background but not execution structure; raw logs record execution but lack stable anchors. We therefore use two complementary substrates and query each with a graph-native distance: a **breadth semantic graph** for broad, low-variance background and a **depth causal graph** for short, reproducible procedure chains. See Figure 2 for the overall workflow.

## 3.1 Dual-Graph Collaboration

**Motivation.** Scientific reasoning systems draw on two distinct sources: **(i) static background knowledge**, including entities, definitions, equations, and cross-document evidence that remain stable across queries and are typically derived from documents or websites; and **(ii) procedural knowledge**, consisting of transient, stepwise traces generated through *search–parse–compute–verify* interactions, such as tool calls, intermediate artifacts, and validator outcomes. Both are essentia, the first anchors terms and reduces hallucination, and the second mirrors the thinking process and supports reasoning. However, most methods focus on static semantics and treat procedural signals as loose context, or even omit its semantics, hence leading to the suboptimal results.

**Definition 1** (Breadth Semantic Graph by Static Background). The Breadth graph is $G^B = (V^B, E^B, s_B)$. Nodes $V^B$ are canonical entities/terms, paragraph or table spans, and formula symbols. Edges $E^B$ encode lightweight semantic/evidential relations (`mentions`, `defines`, `aliases`, `cites`, `supports`, `derived_from`). Each edge $e$ carries a normalized confidence $s_B(e) \in (0, 1]$ summarizing extraction reliability and cross-source support. This layer provides stable anchors and multi-hop background structure; it deliberately avoids procedural detail.

**Definition 2** (Depth Causal Graph by Procedural Background). The Depth graph is $G^D = (V^D, E^D, s_D)$. Nodes $V^D$ abstract execution provenance into *Action* (operator/tool with parameters and an environment signature), *Artifact/Result* (intermediate or final quantities), and *Validator* (unit/equation/consistency checks). Directed edges $E^D$ capture stepwise dependency observed in logs: `consumes`(*Artifact → Action*), `produces`(*Action → Artifact*), `verified_by`(*Artifact → Validator*), and carryover when an output becomes a downstream input. An edge is admitted only if typing, units, and temporal order are coherent; admitted edges receive a single confidence $s_D(e) \in (0, 1]$ derived from validator success and repeatability across retries/branches. This layer encodes reproducible, checkable chains rather than textual co-occurrence.

**Entity/Relation Extraction for the Breadth Semantic Graph.** For each span of background knowledge retrieved in the log, we feed the raw sentence or paragraph into an LLM with a fixed extraction prompt that asks it to enumerate all salient entities (such as concepts, numerical values, and units) and the semantic relations between them (for example, "is defined as", "refers to", or "cites"). The model's output is a small set of triple-like records, which we normalize into canonical entity mentions and relation types. Each entity becomes a node in $V^B$, and each relation becomes a typed edge in $E^B$. Because the same concept may be extracted from multiple passages, we aggregate these signals and assign every edge $e$ a confidence score $s_B(e)$ that reflects the reliability and consistency of the extraction. Running this template-driven procedure over all retrieved snippets systematically turns unstructured background text into a Breadth Semantic Graph that covers key concepts and their cross-document links.

**Action/Artifact Parsing for the Depth Causal Graph.** For every record in the execution log, including agent decisions, tool invocations, and explicit verification steps, we apply a structured parsing prompt that asks the LLM to list: (i) the action or tool and its parameters, (ii) the input artifacts, (iii) the output artifacts, and (iv) any validations performed. From this parsed output, we instantiate a single *Action* node and several *Artifact* nodes, and connect them with directed `consumes` and `produces` edges. If the log entry contains a consistency or unit check, we additionally create a *Validator* node and attach `verified_by` edges from the corresponding artifacts. An edge is admitted into $E^D$ only when basic type, unit, and temporal constraints inferred from the log are satisfied, and its confidence $s_D(e)$ encodes how strongly these checks succeed. This process transforms raw, sequential logs into a coherent Depth Causal Graph that captures reproducible chains of computation rather than opaque text traces.

**Semantic Retrieval on the Two Graphs.** Formally, $f(\cdot)$ serves as the encoder of query $q$, while $g_B(\cdot)$ and $g_D(\cdot)$ serve as the node and edge encoders for the Breadth and Depth graph, respectively. We define *semantic* retrieval scores as follows.

**Breadth similarity (semantic anchoring).** We score a background node $v \in V^B$ by comparing the query to a *neighborhood–smoothed* representation of $v$, the node embedding lightly averaged with its immediate neighbors using edge confidences as weights. The breadth score is a single cosine:

$$S_B(v \mid q) = \cos\big(f(q), \bar{g}_B(v)\big). \tag{1}$$

Intuitively, $\bar{g}_B(v)$ suppresses spurious matches to isolated nodes and rewards terms that are semantically close to the query and supported by evidence. This one-hop smoothing avoids multi-step diffusion and hyperparameters, yet retains a topology-aware bias robust to noise extractions.

**Depth similarity (order- and type-aware).** Cosine on a path embedding ignores ordering and typed constraints. We instead compare the *operation sequence* implied by the query to that of a short admissible chain. Let $O(q)$ be the sequence of required typed operations extracted from $q$ (e.g., search→parse→compute→verify), and let $O(p)$ be the action/validator sequence on path $p$ (passing type/unit/time gates). Define $\text{LCS}^\dagger(O(q), O(p))$ as the longest common subsequence that only counts matches with compatible types/units. With a simple path reliability $R(p) = \left( \prod_{e \in p} s_D(e) \right)^\tau$ ($\tau \in (0, 1]$), we use:

$$S_D(t \mid q) = \max_{p \in \mathcal{P}_{\leq L}(t)} R(p) \cdot \frac{\text{LCS}^\dagger(O(q), O(p))}{|O(q)|}. \tag{2}$$

This single-score criterion is simple, auditable, and efficient (dynamic programming on short sequences). It favors targets supported by brief, reliable chains that *respect the query's procedural order and typing*, without relying on embedding cosines.

Equation 1 provides a one–hop, neighborhood–smoothed *semantic* score on the Breadth graph: it favors nodes that are textually close to the query while being supported by nearby evidence, yielding a stable, topology–aware anchor without multi-step diffusion. In contrast, Equation 2 supplies an *order- and type-aware* process score on the Depth graph: a target is relevant only if there exists a short admissible chain whose action/validator sequence (and units/types) matches the query's required operation pattern, with reliability encouraging brief, high-confidence procedures. Taken together, these complementary signals capture both *where the facts live* and *how the result is produced*, producing compact, auditable evidence that improves downstream reasoning.

## 3.2 DUAL-CHANNEL ENTROPY AGGREGATION

Given a query $q$ and a finite answer set $\mathcal{A}$, our goal is to select $a^\star \in \mathcal{A}$ while returning a compact, checkable evidence chain. We have obtained the content after two graph retrievals, then we aggregate the information as follows.

**Path Scoring with Drift Control** A path $p$ in $\mathcal{G}^B$ (obtained by path-constrained search) receives the log-additive score:

$$S_B(p \mid q) = \sum_{e \in p} \log w_e^B - \lambda_{\text{off}} \cdot \text{Offtopic}(p), \tag{3}$$

where $\text{Offtopic}(p) \geq 0$ penalizes topical drift from $q$ accumulated along $p$ and $\lambda_{\text{off}} \geq 0$ controls the strength. An analogous score $S_D(p \mid q)$ is computed on $\mathcal{G}^D$, incorporating edge direction, type and temporal consistency.

**From Paths to Per–Channel Answer Distributions** Let $\mathcal{P}_B(a)$ and $\mathcal{P}_D(a)$ denote the sets of breadth/causal paths that support answer $a$ (possibly filtered/verified by an LLM over their stitched contexts). We map path scores to per–channel answer distributions via log-sum-exp aggregation:

$$P_B(a \mid q) = \frac{\sum_{p \in \mathcal{P}_B(a)} \exp\big(S_B(p \mid q)\big)}{\sum_{a' \in \mathcal{A}} \sum_{p \in \mathcal{P}_B(a')} \exp\big(S_B(p \mid q)\big)}, P_D(a \mid q) = \frac{\sum_{p \in \mathcal{P}_D(a)} \exp\big(S_D(p \mid q)\big)}{\sum_{a' \in \mathcal{A}} \sum_{p \in \mathcal{P}_D(a')} \exp\big(S_D(p \mid q)\big)}. \tag{4}$$

**Entropy–Driven Log–Linear Fusion** We quantify each channel's certainty using Shannon entropy,

$$H_B = -\sum_{a \in \mathcal{A}} P_B(a \mid q) \log P_B(a \mid q), \qquad H_D = -\sum_{a \in \mathcal{A}} P_D(a \mid q) \log P_D(a \mid q), \tag{5}$$

and fuse the channels in log-space with a data–dependent gate:

$$P(a \mid q) = \text{softmax}\Big(\alpha(H) \cdot \log P_D(a \mid q) + \big(1 - \alpha(H)\big) \cdot \log P_B(a \mid q)\Big), \tag{6}$$

$$\alpha(H) = \frac{\exp(-H_D)}{\exp(-H_D) + \exp(-H_B)} \in [0, 1]. \tag{7}$$

Intuitively, the more peaked (certain) channel receives the larger weight; diffuse (high-entropy) evidence is down-weighted. When both channels are confident and consistent, their signals are amplified by the fusion.

**Global Calibration** We further calibrate the fused distribution to discourage overconfidence under global uncertainty:

$$\tilde{P}(a \mid q) = \text{softmax}\left(\tfrac{1}{\gamma} \log P(a \mid q) - \beta \cdot (H_B + H_D)\right), \tag{8}$$

with temperature $\gamma > 0$ (smaller $\gamma$ sharpens) and penalty $\beta \geq 0$.

**Answer Selection and Minimal Evidence Chain** The final prediction is the MAP answer

$$a^\star = \arg \max_{a \in \mathcal{A}} \tilde{P}(a \mid q). \tag{9}$$

To return a verifiable rationale, we extract a *minimal evidence chain* on $\mathcal{G}^D \cup \mathcal{G}^B$ supporting $a^\star$. Let $\Delta_e$ denote the marginal contribution of edge $e$ to $\tilde{P}(a^\star \mid q)$, measured as the drop in $\tilde{P}(a^\star \mid q)$ when removing $e$ (leave-one-out). We greedily prune edges in ascending $\Delta_e$ until the cumulative drop exceeds a threshold $\delta > 0$, and report the remaining path. Because $\mathcal{G}^D$ encodes directed, typed and temporal constraints while $\mathcal{G}^B$ provides cross-document coverage, the resulting chain is simultaneously *deep* and *broad*.

Eqs. 6–8 make the decision rule *evidence-adaptive*: a high-entropy channel cannot dominate the prediction, while agreement between confident channels is explicitly amplified. The off-topic penalty in Eq. 3 curbs topical drift during breadth exploration. The path-constrained extraction provides a compact, checkable rationale for $a^\star$, aligning the final output with graph-grounded evidence. More theoretical analysis can be found in Appendix A.

# 4 EXPERIMENT

In this section, the experimental setup is first introduced. Subsequently, the improvements of the proposed method over the baselines are demonstrated. Then, comparisons with existing methods, ablation studies on different components, and more targeted analyses are also presented.

## 4.1 EXPERIMENTAL DETAILS

**Baseline and Setting.** In this study, we adopt InternAgent (Team et al., 2025) as the baseline. And we collected log files generated during InternAgent problem-solving process and subsequently cleaned them, thereby providing the foundation for graph construction. The results of QwQ-32B, DeepSeek-R1-671B, and WebThinker-32B-RL are obtained from Li et al. (2025). The results of SFR-DR-20B (Nguyen et al., 2025) and X-Masters (Chai et al., 2025) are drawn from their papers. Results for OpenAI Deep Research (OpenAI, 2025c) and Gemini Deep Research (DeepMind, 2024) are taken from their respective technical reports. For Qwen3-235B-A22B-Instruct (Yang et al., 2025), GPT-5 (OpenAI, 2025a), and o4-mini (OpenAI, 2025b), we conducted direct evaluations through the API (see Appendix B.2 for prompt settings), with the temperature set to 0.0 and no restriction on the maximum token limit.

**Benchmark. GAIA** (Mialon et al., 2023) is a benchmark for general-purpose AI assistants comprising 466 real-world, information-seeking questions that require multi-step reasoning, multimodal understanding, web browsing, and tool use. Its tasks are designed to be straightforward for humans yet challenging for current models, with closed-form answers and a held-out subset used to support robust, leaderboard-style evaluation of agentic systems. Our results are based on its 165-question validation set. **Google-Proof Q&A (GPQA)** (Rein et al., 2024) is a graduate-level benchmark of 448 expert-written multiple-choice questions in biology, chemistry, and physics, designed to test advanced scientific reasoning. We adopt its GPQA-Diamond subset (198 questions), which was curated to include only items unanimously agreed upon by domain experts but often misanswered by non-experts, ensuring both reliability and difficulty. **Humanity's Last Exam (HLE)** (Phan et al., 2025) is a multimodal benchmark of 2,500 expert-curated, closed-form questions across eight domains. It assesses advanced reasoning through multiple-choice and exact-match tasks, comprising 2,158 text-only and 342 text–image items, thereby enabling rigorous evaluation across modalities.

Table 1: Comparison on **HLE** and **GPQA**. We report per-subset accuracy for the baseline InternAgent and DualResearch under two settings. Improvements over the baselines are highlighted in red.

| Data Setting | Model | Method | Accuracy in each subset (%) ↑ | | | | | | | | |
|---|---|---|---|---|---|---|---|---|---|---|---|
| | | | Math | Bio/Med | CS/AI | Physics | Human. | Chem. | Engineer. | Other | Avg. |
| HLE Text-Only | Qwen3-235B -A22B-Instruct | InternAgent | 13.5 | 15.3 | 11.6 | 6.9 | 16.1 | 8.9 | 15.6 | 17.6 | 13.3 |
| | | DualResearch | 16.9 | 18.9 | 13.9 | 10.9 | 19.7 | 21.8 | 21.9 | 19.9 | 17.1 |
| | | Improvement | ↑3.4 | ↑3.6 | ↑2.3 | ↑4.0 | ↑3.6 | ↑12.9 | ↑6.3 | ↑2.3 | ↑3.8 |
| | o4-mini | InternAgent | 23.5 | 18.9 | 13.9 | 17.3 | 21.6 | 21.6 | 18.8 | 25.0 | 21.3 |
| | | DualResearch | 31.3 | 27.0 | 26.8 | 24.3 | 29.0 | 25.7 | 28.1 | 30.1 | 29.0 |
| | | Improvement | ↑7.8 | ↑8.1 | ↑12.9 | ↑7.0 | ↑7.4 | ↑4.1 | ↑9.3 | ↑5.1 | ↑7.7 |
| HLE All-Set | Qwen3-235B -A22B-Instruct | InternAgent | 13.0 | 12.5 | 10.8 | 7.4 | 14.2 | 7.9 | 9.0 | 13.7 | 11.9 |
| | | DualResearch | 16.2 | 15.0 | 12.8 | 9.6 | 17.4 | 13.3 | 12.6 | 15.0 | 14.8 |
| | | Improvement | ↑3.2 | ↑2.5 | ↑2.0 | ↑2.2 | ↑3.2 | ↑5.4 | ↑3.6 | ↑1.3 | ↑2.9 |
| | o4-mini | InternAgent | 23.5 | 18.9 | 17.4 | 15.7 | 19.2 | 18.2 | 16.2 | 20.6 | 20.1 |
| | | DualResearch | 32.3 | 27.9 | 27.0 | 23.5 | 27.4 | 21.2 | 21.6 | 25.8 | 27.8 |
| | | Improvement | ↑8.8 | ↑9.0 | ↑9.6 | ↑7.8 | ↑8.2 | ↑3.0 | ↑5.4 | ↑5.2 | ↑7.7 |
| GPQA -diamond | o4-mini | InternAgent | - | 73.68 | - | 94.19 | - | - | 70.97 | - | 81.31 |
| | | DualResearch | - | 84.21 | - | 96.51 | - | - | 79.57 | - | 87.37 |
| | | Improvement | - | ↑10.53 | - | ↑2.32 | - | - | ↑8.60 | - | ↑6.06 |

## 4.2 COMPARISON WITH BASELINE METHOD

We quantify the improvements of DualResearch over InternAgent across datasets and backbones. As shown in Table 1, **On the HLE Text-Only**, DualResearch demonstrates significant improvements over the baseline InternAgent across two different models. Specifically, it achieves a 12.9% increase in Chemistry with Qwen3 and in CS/AI with o4-mini, along with average accuracy gains of 3.8% and 7.7%, respectively. Similar improvements remain evident **on the HLE All-Set**, where average accuracy increases by 2.9% and 7.7%. **On GPQA**, using o4-mini as the backbone model, the largest improvement is observed in Biology, with an increase of 10.53%. The overall average accuracy also rises by 6.06%. These stable improvements substantiate that, after reusing InternAgent's solution logs, DualResearch consistently amplifies effective evidence while suppressing irrelevant information, leading to sustained and reproducible performance gains.

In addition, we reproduced the results of X-Masters on Bio/Med within HLE Text-Only. Based on its logs, DualResearch achieving an improvement of 4.9% (see Appendix B.1 for details).

## 4.3 COMPARISON BETWEEN DUALRESEARCH AND EXISTING WORK

In this section, we compare the proposed method with existing approaches. As illustrated in Table 2, DualResearch remains highly competitive among all evaluated systems. On the HLE Text-Only task, DualResearch achieves an average accuracy of 29.0%, below Tongyi-DeepResearch and X-Masters. Note that DualResearch itself is not a standalone deep-research agent, it is a post-hoc module that operates on the execution logs of existing systems. Nevertheless, it substantially improves over the corresponding single-turn baselines and the original InternAgent results under the same backbone models, and attains the best or second-best performance in most disciplines. Moreover, X-Masters further relies on a multi-channel majority-voting strategy that markedly increases token usage, whereas DualResearch can be easily plugged into different agents with minimal overhead. These observations indicate that DualResearch possesses good transferability and plug-and-play capability, making it a general enhancement module for current and future deep-research systems rather than a competing end-to-end solution.

In the All-Set setting, the average accuracy was 27.8%, outperforming the second-best, Gemini Deep Research, by 0.9%. Here, it ranked first in seven out of eight fields, placing second in Engineering. Compared with agent baselines using the same backbone model, our method consistently demonstrated significant improvements. On o4-mini, DualResearch outperformed InternAgent by 7.3% in the Text-Only setting and by 7.3% in the All-Set setting.

Table 2: Comparison on **HLE**. The best results are **bolded**, and the second-best results are underlined. Methods marked † are direct model evaluations, those marked ‡ are agent-based. Here, "Qwen3-235B" denotes "Qwen3-235B-A22B-Instruct."

| | Method | Accuracy in each subset (%) ↑ | | | | | | | | |
|---|---|---|---|---|---|---|---|---|---|---|
| | | Math | Bio/Med | CS/AI | Physics | Human. | Chem. | Engineer. | Other | Avg. |
| Text-Only | QwQ-32B † | 12.6 | 14.0 | 7.9 | 4.0 | 6.0 | 13.3 | 5.3 | 4.4 | 9.6 |
| | DeepSeek-R1-671B † | 9.3 | 8.6 | 7.4 | 5.8 | 11.0 | 5.6 | 10.3 | 7.5 | 8.6 |
| | Qwen3-235B † | 11.4 | 9.0 | 8.5 | 5.5 | 8.3 | 4.9 | 7.8 | 6.3 | 9.2 |
| | o4-mini † | 19.7 | 9.9 | 13.4 | 13.4 | 9.8 | 6.9 | 9.4 | 6.8 | 14.5 |
| | GPT-5 † | 31.3 | 21.2 | 25.5 | 23.3 | 21.8 | 18.8 | 10.9 | 19.3 | 25.9 |
| | WebThinker-32B-RL ‡ | 16.7 | 25.6 | 2.0 | 12.7 | 18.0 | **26.7** | 15.8 | 15.6 | 15.8 |
| | SFR-DR-20B ‡ | - | - | - | - | - | - | - | - | 28.7 |
| | X-Masters ‡ | **38.5** | **27.6** | 22.5 | 24.1 | **33.2** | 26.1 | 23.4 | 29.0 | 32.1 |
| | Tongy ‡ | - | - | - | - | - | - | - | - | **32.9** |
| | Kimi-Research ‡ | - | - | - | - | - | - | - | - | 26.9 |
| | InternAgent (Qwen3-235B) ‡ | 13.5 | 15.3 | 11.6 | 6.9 | 16.1 | 8.9 | 15.6 | 17.6 | 13.3 |
| | InternAgent (o4-mini) ‡ | 23.5 | 18.9 | 13.9 | 17.3 | 21.6 | 21.6 | 18.8 | 25.0 | 21.3 |
| | DualResearch (Qwen3-235B ‡) | 16.9 | 18.9 | 13.9 | 10.9 | 19.7 | 21.8 | 21.9 | 19.9 | 17.1 |
| | DualResearch (o4-mini) ‡ | 31.3 | 27.0 | **26.8** | **24.3** | 29.0 | 25.7 | **28.1** | **30.1** | 29.0 |
| All-Set | Qwen3-235B † | 11.1 | 7.9 | 8.3 | 6.1 | 7.8 | 5.5 | 7.2 | 5.2 | 8.6 |
| | o4-mini † | 19.0 | 11.4 | 12.9 | 12.6 | 9.1 | 12.7 | 12.6 | 6.9 | 14.3 |
| | GPT-5 † | 31.0 | 22.1 | 24.9 | 21.7 | 20.6 | 16.4 | 14.4 | 18.0 | 24.8 |
| | OpenAI Deep Research ‡ | - | - | - | - | - | - | - | - | 26.6 |
| | Gemini Deep Research ‡ | - | - | - | - | - | - | - | - | 26.9 |
| | InternAgent (Qwen3-235B) ‡ | 13.0 | 12.5 | 10.8 | 7.4 | 14.2 | 7.9 | 9.0 | 13.7 | 11.9 |
| | InternAgent (o4-mini) ‡ | 23.5 | 18.9 | 17.4 | 15.7 | 19.2 | 18.2 | 16.2 | 20.6 | 20.1 |
| | DualResearch (Qwen3-235B ‡) | 16.2 | 15.0 | 12.8 | 9.6 | 17.4 | 13.3 | 12.6 | 15.0 | 14.8 |
| | DualResearch (o4-mini) ‡ | **32.3** | **27.9** | **27.0** | **23.5** | **27.4** | **21.2** | 21.6 | **25.8** | **27.8** |

Table 3 shows comparison on GPQA, DualResearch likewise exhibited superior performance, achieving an average accuracy of 87.37%, which is 2.02% higher than direct inference with GPT-5. By subset, on Bio, the result tied with the best; on Chem, the improvement was most pronounced at 3.23%; and on Phys, it further led by 1.16%. In addition, on the same backbone model o4-mini, our average accuracy is 6.06% higher than that of InternAgent. Overall, our method distills the logs of DeepResearch into declarative and procedural knowledge. This reduces the solution space, decreases uncertainty, and ultimately delivers more stable improvements.

Table 4 reports the comparison on GAIA. DualResearch again achieves the best overall performance, obtaining an average accuracy of 71.10%, which is 3.74% higher than OpenAI Deep Research. By difficulty level, it reaches 84.95% and 73.58% on Level-1 and Level-2, surpassing OpenAI Deep Research by 10.66% and 4.52%, respectively. In the most challenging Level-3 split, it ranks second while still outperforming InternAgent by 10.20%.

Table 3: Comparison on **GPQA**. The best results are **bolded**, and the second-best results are underlined. Methods marked † are direct model evaluations, those marked ‡ are agent-based.

| Method | Accuracy in each subset (%) ↑ | | | |
|---|---|---|---|---|
| | Bio | Chem | Phys | Avg. |
| DeepSeek-R1-671B † | 63.16 | 76.34 | 91.86 | 82.32 |
| o4-mini † | 78.95 | 63.44 | 94.19 | 78.28 |
| GPT-5 † | **84.21** | 76.34 | 95.35 | 85.35 |
| WebThinker-32B-RL ‡ | 78.90 | 50.50 | 90.70 | 70.70 |
| InternAgent (o4-mini) ‡ | 73.68 | 70.97 | 94.19 | 81.31 |
| DualResearch (o4-mini) ‡ | **84.21** | **79.57** | **96.51** | **87.37** |

Table 4: Comparison on **GAIA**. The best results are **bolded**, and the second-best results are underlined. Methods marked † are direct model evaluations, those marked ‡ are agent-based.

| Method | Accuracy in each subset (%) ↑ | | | |
|---|---|---|---|---|
| | Level-1 | Level-2 | Level-3 | Avg. |
| Qwen3-235B † | 15.09 | 3.49 | 3.84 | 6.67 |
| o4-mini † | 28.30 | 12.79 | 7.69 | 16.97 |
| DeepSeek-R1-671B † | 33.96 | 13.95 | 3.84 | 18.74 |
| OpenAI Deep Research ‡ | 74.29 | 69.06 | **47.60** | 67.36 |
| InternAgent (o4-mini) ‡ | 78.49 | 69.18 | 26.53 | 65.12 |
| DualResearch (o4-mini) ‡ | **84.95** | **73.58** | 36.73 | **71.10** |

## 4.4 ABLATION STUDY

In this section, we conduct an ablation study on the key components based on o4-mini to verify the contribution and complementarity of each submodule. Table 5 reports the results on HLE and GPQA. First, it can be observed that single-source information already brings consistent gains: on HLE, Breadth and Depth improve performance by 3.5% and 3.4%, respectively; on GPQA, Depth yields an improvement of 1.52%. Second, directly concatenating Breadth and Depth introduces conflicts and noise, resulting in only a 1.8% improvement over the HLE baseline and even a 0.51% decrease on GPQA. This indicates that unconstrained fusion of the two types of evidence enlarges the solution space and amplifies uncertainty. Finally, with the introduction of entropy-based

Table 5: Ablation of components with o4-mini. The best results are **bolded**.

| Components | | | Datasets | |
|---|---|---|---|---|
| Breadth | Depth | Aggregation | HLE | GPQA |
| ✗ | ✗ | ✗ | 20.1 | 81.31 |
| ✔ | ✗ | ✗ | 23.6 | 81.31 |
| ✗ | ✔ | ✗ | 23.5 | 82.83 |
| ✔ | ✔ | ✗ | 21.9 | 80.80 |
| ✔ | ✔ | ✔ | **27.8** | **87.37** |

aggregation, the model achieves the best results, reaching 27.8% on HLE and 87.37% on GPQA. This aligns with our original design motivation: by hierarchically modeling declarative and procedural evidence from logs, and employing entropy-driven aggregation to adaptively select high-confidence and low-redundancy information, we avoid the expansion of the search space caused by naive concatenation, thereby improving performance on both benchmarks simultaneously.

## 4.5 ANALYSIS & VISUALIZATION

**Case Study.** Figure 3 highlights the differences between InternAgent and DualResearch in evidence

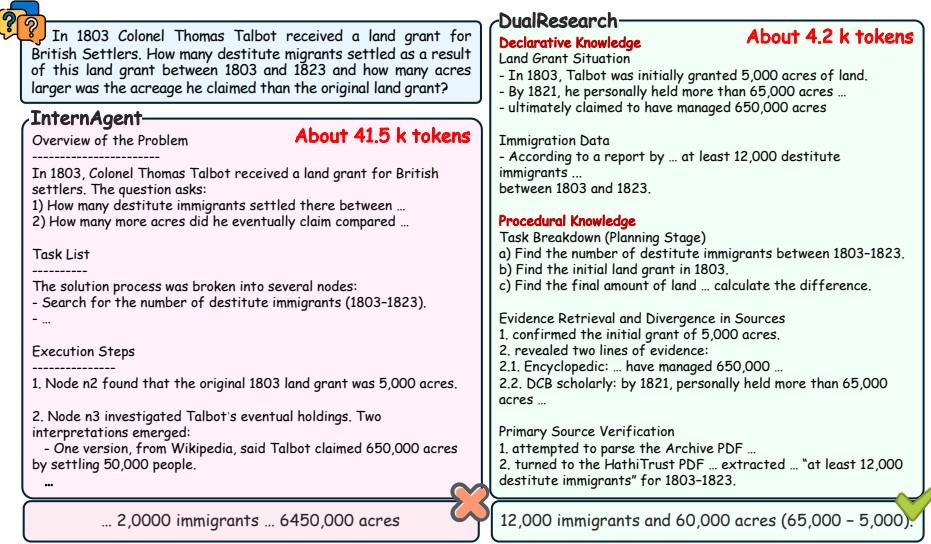

Figure 3: Case study of a historical query comparing InternAgent and DualResearch.

management and constraint modeling. InternAgent generated a 41.5k-token log that repeatedly conflated contradictory cues, for example, equating "claimed" with "managed" and misinterpreting "at least 12,000" as a larger value. The long record accumulated noise rather than certainty.

In contrast, DualResearch distilled the same log into structured evidence: (1) declarative facts, such as the grant of 5,000 acres, 12,000 immigrants (1803–1823), and over 65,000 acres personally held by 1821; and (2) procedural steps, including constraining time, normalizing units, and restricting "claimed" to personal holdings. It then applied entropy-gated selection, down-weighting the 650,000-acre claim as uncertain and retaining the 65,000-acre figure as reliable. With a compact 4.2k-token graph, it produced the correct result: 12,000 immigrants and a 60,000-acre difference. These findings show that layered representations and uncertainty-driven evidence selection improve both interpretive consistency and computational reliability.

**Signal and Subject Graph Construction Strategies.** Figure 4 shows that subject-level multi-sample graph aggregation outperforms the Signal baseline in Bio/Med, Chemistry, Engineering,

Physics, and Mathematics, benefiting from strong ontological consistency and standardized notation. By reusing entities and relations across samples, aggregation yields denser graphs that enhance long-chain reasoning efficiency and verification accuracy. Conversely, in domains with greater heterogeneity, such as Other, Humanities, and CS/AI, Signal's single-sample graphs prove more effective, as direct merging may introduce contradictory or weakly related edges, amplifying noise and undermining causal coherence. The small gap indicates DualResearch adapts, strengthening reusable edges in coherent domains and suppressing weak or conflicting ones in heterogeneous settings, balancing overall gains with domain-specific contrasts.

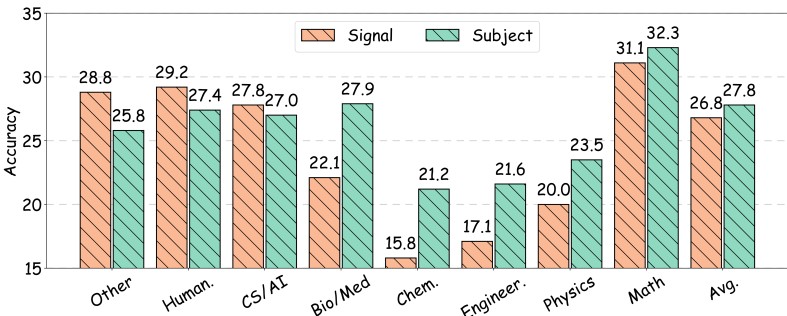

Figure 4: Accuracy comparison between two graph construction strategies across disciplines. Signal denotes single-sample graph construction, where each instance independently forms a knowledge graph. Subject denotes subject-level multi-sample graph aggregation, where multiple instances within the same discipline are merged into a unified graph.

## 5 CONCLUSION

In this paper, we have introduced DualResearch, a dual-graph retrieval and fusion framework designed for tool-intensive scientific reasoning tasks. The method constructs both a breadth knowledge graph and a depth process graph: the former provides stable background knowledge, while the latter captures executable reasoning processes. In extensive experiments, DualResearch can directly leverage the problem-solving trajectories of existing deep research frameworks to yield significant improvements in both accuracy and verifiability. On scientific benchmarks such as HLE and GPQA, our approach substantially outperforms strong baselines and maintains competitiveness across multidimensional evaluation metrics.

In the future, we plan to extend DualResearch to multimodal scientific reasoning by integrating diverse sources of evidence such as figures, tables, and experimental data. We believe that this direction will further advance the breadth and depth of automated scientific discovery.

# 6 ETHICS STATEMENT

This article does not involve research on human subjects, practices of dataset release, insights, methods, or applications with potential harm, or other related ethical issues.

# 7 REPRODUCIBILITY STATEMENT

We have made every effort to ensure the reproducibility of our research results. The implementation details of the proposed DualResearch framework, including graph construction rules and entropy-gated aggregation, are described in the main text and further elaborated in the appendix. For theoretical contributions, all assumptions are explicitly stated, and complete proofs of the related claims are provided in the appendix. For empirical evaluations, we used publicly available datasets, with the processing pipeline and experimental settings documented in the supplementary materials. All relevant code for this study will be released on GitHub upon acceptance of the paper.

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

## A  THEORY

### A.1  RATIONALE AND SCOPE

Complex, tool-intensive queries expose two complementary information channels: a *breadth* channel that aggregates stable background evidence across documents, and a *depth* channel that captures instance-specific procedural traces (tools, intermediates, validations). These channels differ systematically in structure (undirected relatedness vs. directed, typed causality) and uncertainty (low-variance coverage vs. path-dependent reliability). In practice, which channel is preferable is *instance-dependent*: breadth can dominate when background suffices, while depth is decisive when procedural constraints determine correctness. Collapsing both into a single latent space fixes an inductive bias that cannot adapt per query and invites ad-hoc fusion without guarantees.

We therefore cast retrieval as a *mixture-of-experts* problem under a proper scoring rule (log-loss), where breadth and depth provide probabilistic posteriors and a gate arbitrates their contributions. Our gate is *uncertainty-aware* via Shannon entropy, a choice that is both operational and theoretically convenient: (i) geometric (log-linear) mixtures admit pointwise upper bounds by convex combinations of expert losses; and (ii) under mild, testable entropy–loss calibration, lower entropy predicts lower conditional loss, so an entropy gate approximates the oracle that selects the better expert per instance. The theorem below formalizes this intuition as an *oracle inequality* with a measurable gating regret that vanishes when the entropy ordering matches the conditional-loss ordering. This yields a principled explanation of why dual-graph fusion can strictly outperform any single graph in expectation, while remaining faithful to each channel's inductive bias.

**Definition 3** (Dual-graph posteriors and entropy-gated fusion)**.** Let $\mathcal{A}$ be a finite answer set and $(q, y) \sim \mathcal{D}$ be query–label pairs. Two layer-native predictors produce posteriors $P_B(\cdot \mid q)$ (Breadth graph) and $P_D(\cdot \mid q)$ (Depth graph). Define their Shannon entropies $H_B(q) = -\sum_{a \in \mathcal{A}} P_B(a \mid q) \log P_B(a \mid q)$ and $H_D(q)$ analogously, and per-example log-losses $\ell_B(q, y) = -\log P_B(y \mid q)$, $\ell_D(q, y) = -\log P_D(y \mid q)$. The fused posterior is the geometric (log-linear) mixture

$$P_F(a \mid q) \;\propto\; P_B(a \mid q)^{\,1-\alpha(H)} \, P_D(a \mid q)^{\,\alpha(H)}, \quad \alpha(H) \;=\; \frac{e^{-H_D(q)}}{e^{-H_D(q)} + e^{-H_B(q)}} \in [0, 1],$$

with population (expected) log-loss risk $\mathcal{R}(P) = \mathbb{E}_{(q,y) \sim \mathcal{D}}\big[ -\log P(y \mid q)\big]$. We say channel $i \in \{B, D\}$ is *entropy–loss calibrated* if there exists a nondecreasing $\phi_i : \mathbb{R}_+ \to \mathbb{R}_+$ such that $\mathbb{E}[\ell_i(q, y) \mid H_i(q) = h] = \phi_i(h)$ almost everywhere.

**Theorem 1** (Generalization advantage of entropy-gated dual-graph fusion)**.** Under the setting above, for any $(q, y)$ and any fixed $\alpha \in [0, 1]$,

$$-\log P_F(y \mid q) \;\leq\; (1 - \alpha)\, \ell_B(q, y) + \alpha\, \ell_D(q, y).$$

Consequently, for the entropy gate $\alpha(H)$,

$$\mathcal{R}(P_F) \;\leq\; \mathbb{E}\big[\min\{\mathbb{E}[\ell_B \mid H], \mathbb{E}[\ell_D \mid H]\}\big] + \mathcal{E}_{\text{gate}}, \qquad \mathcal{E}_{\text{gate}} = \mathbb{E}[|\Delta(H)| \cdot |\alpha(H) - \alpha^{\star}(H)|], \tag{10}$$

where $H = (H_B, H_D)$, $\Delta(H) = \mathbb{E}[\ell_D \mid H] - \mathbb{E}[\ell_B \mid H]$, and $\alpha^{\star}(H) = \mathbb{I}\{\Delta(H) < 0\}$ is the oracle gate. If both channels are entropy–loss calibrated and the *sign-consistency* condition holds almost surely,

$$\text{sign}\big(\Delta(H)\big) \;=\; \text{sign}\big(H_D - H_B\big),$$

then $\alpha(H) = \alpha^{\star}(H)$ almost surely, $\mathcal{E}_{\text{gate}} = 0$, and

$$\mathcal{R}(P_F) \;\leq\; \mathbb{E}\big[\min\{\mathbb{E}[\ell_B \mid H], \mathbb{E}[\ell_D \mid H]\}\big] \;\leq\; \min\{\mathcal{R}(P_B), \mathcal{R}(P_D)\}. \tag{11}$$

Thus the entropy-gated dual-graph fusion generalizes at least as well as the better single graph and strictly better whenever the better channel varies across queries.

*Proof.* **Step 1: Pointwise bound for geometric mixtures.** For any $\alpha \in [0, 1]$, the fused posterior can be written as $P_F(y \mid q) = \frac{P_B(y|q)^{1-\alpha} P_D(y|q)^{\alpha}}{Z}$ with $Z = \sum_a P_B(a \mid q)^{1-\alpha} P_D(a \mid q)^{\alpha}$. By Hölder's inequality (generalized AM–GM), $\sum_a x_a^{1-\alpha} y_a^{\alpha} \leq (\sum_a x_a)^{1-\alpha}(\sum_a y_a)^{\alpha}$ for $x_a, y_a \geq 0$. Taking $x_a = P_B(a \mid q)$, $y_a = P_D(a \mid q)$ (each sums to 1), we have $Z \leq 1$ and hence $\log Z \leq 0$. Therefore, we have

$$-\log P_F(y \mid q) = -(1-\alpha) \log P_B(y \mid q) - \alpha \log P_D(y \mid q) + \log Z \leq (1-\alpha)\ell_B(q, y) + \alpha \ell_D(q, y).$$

**Step 2: Expectation and oracle decomposition.** Taking expectations and letting the gate be data-dependent $\alpha(H)$, we have

$$\mathcal{R}(P_F) \leq \mathbb{E}[\ell_B] + \mathbb{E}[\alpha(H) \cdot (\ell_D - \ell_B)] = \mathbb{E}\big[\mathbb{E}[\ell_B \mid H] + \alpha(H)\big(\mathbb{E}[\ell_D \mid H] - \mathbb{E}[\ell_B \mid H]\big)\big].$$

Let $\Delta(H) = \mathbb{E}[\ell_D \mid H] - \mathbb{E}[\ell_B \mid H]$ and define the oracle gate $\alpha^\star(H) = \mathbb{I}\{\Delta(H) < 0\}$. Then

$$\mathbb{E}[\ell_B \mid H] + \alpha(H)\Delta(H) = \min\{\mathbb{E}[\ell_B \mid H], \mathbb{E}[\ell_D \mid H]\} + (\alpha(H) - \alpha^\star(H))\Delta(H).$$

Taking outer expectations and applying $|\mathbb{E}[X]| \leq \mathbb{E}[|X|]$ yields equation 10.

**Step 3: Calibration implies sign-consistent gating.** If channels are entropy–loss calibrated, there exist nondecreasing $\phi_B, \phi_D$ with $\mathbb{E}[\ell_i \mid H_i = h] = \phi_i(h)$. Hence $\Delta(H) = \phi_D(H_D) - \phi_B(H_B)$ and $\text{sign}(\Delta(H)) = \text{sign}(H_D - H_B)$ whenever $\phi_i$ are strictly increasing. Our entropy gate $\alpha(H) = \frac{e^{-H_D}}{e^{-H_D} + e^{-H_B}}$ is a strictly increasing function of $-(H_D - H_B)$, thus $\alpha(H) = \alpha^\star(H)$ almost surely under the sign-consistency condition, making $\mathcal{E}_{\text{gate}} = 0$. Finally, Jensen's inequality for the convex function $\min$ gives $\mathbb{E}[\min\{\mathbb{E}[\ell_B \mid H], \mathbb{E}[\ell_D \mid H]\}] \leq \min\{\mathbb{E}[\ell_B], \mathbb{E}[\ell_D]\}$. In this way, we have completed the proof. $\square$

---

**A minimal worked example for depth similarity**

Query. "Consider the song 'All My Loves Are You' as played by Erroll Garner on the 1986 album *Afternoon Of An Elf*. What type of scale does Garner play in the right hand melody between seconds 39 and 43 of the song?" (The album/track info is standard; e.g., *Afternoon of an Elf* lists "All My Loves Are You" and is widely available on streaming.

Answer set. {Chromatic, Major (Ionian), Dorian, Blues}

**What our method expects from the query.**

From the text, the query implies a short, typed procedure:

$O(q)$ = search_track (Audio) → segment_audio (00:39–00:43) → compute_intervals (RH_melody) → verify_scale_pattern (theory_check)

This is exactly the kind of order- and type-aware sequence that our depth score compares against candidate execution chains (Definition 2 and Eq. 2).

Three competing candidate paths from the log and how $S_D$ is computed

We imagine three admissible (typed, unit/time-consistent) chains produced during a tool run. For each chain we compute Eq. (2):

$$S_D(t|q) = \max_{p \in P_{SL}(t)} R(p) \cdot \frac{LCS^\dagger(O(q), O(p))}{|O(q)|}.$$

Path $p_1$ → predicts Chromatic

$O(p_1)$ = [search_track, segment_audio, compute_intervals, verify_scale_pattern]

Reliability: 0.96

Path $p_2$ → predicts Dorian

$O(p_2)$ = [search_track, segment_audio, estimate_key, classify_mode]

Reliability: 0.92, Depth score: 0.46.

Path $p_3$ → predicts Blues

$O(p_3)$ = [search_track, transcribe_chords, classify_blues_scale]

Reliability: 0.94, Depth score: 0.235.

Thus, the order- and type-aware depth similarity strongly favors Chromatic (0.96) over Dorian (0.46) and Blues (0.235). This follows our definition of depth retrieval and typed LCS† exactly.

**Where the breadth channel points**

On the breadth graph we include background nodes like:

"Chromatic scale = sequence of semitone steps" (music-theory node) → matches the query's "what type of scale (melodic pattern) in a 4-s excerpt," and supports an interval-step interpretation. (Definition pages as background.)

In this example, "Chromatic scale" receives a higher breadth score than "Dorian" or "Blues" because the query explicitly targets *semitone-based right-hand melody identification* in a tight time window—semantically closer to the chromatic definition than to modal/hexatonic summaries.

**Fusion and final selection**

Each channel produces a normalized answer distribution (Eq. 4). The entropy-gated fusion then up-weights the sharper (lower-entropy) channel (Eqs. 6–7), amplifying agreement. Here, Depth is very peaked on Chromatic, so $\alpha(H)$ leans toward Depth; the fused posterior's MAP is Chromatic Scale.

**Final answer:** Chromatic Scale.

Figure 5: Example for depth similarity.

# B MORE RESULTS AND DETAILS FOR EXPERIMENTS

In this section, we present the performance improvements achieved by DualResearch when reusing X-Masters logs. We then demonstrate how we configure prompts to enable the LLM to clean logs, as well as evaluate on the HLE and GPQA datasets.

## B.1 IMPROVEMENT IN X-MASTERS

As shown in Table 6, it is evident that after reusing the scientific logs generated by X-Masters during problem solving, DualResearch demonstrates significant improvements. The accuracy increased from 23.9% to 28.8%, yielding a gain of 4.9%. This finding is consistent with the conclusions reported on InternAgent in the main text, providing strong evidence that DualResearch, as a post-processing method for deep research, can deliver stable performance gains.

Table 6: Compared with the baseline X-Masters, DualResearch shows improvements on HLE Text-Only in the Bio/Med domain. Here, X-Masters refers to the originally reported results, while X-Masters denotes the results we obtained from our reproduction. All frameworks employ DeepSeek-R1-671B as the backbone model.

| Methods | X-Masters | X-Masters* | DualResearch | Improvement |
|---|---|---|---|---|
| Acc. in Bio/Med | 27.6 | 23.9 | 28.8 | ↑**4.9** |

### B.2 PROMPT FOR TEST LLMS IN HLE AND GPQA

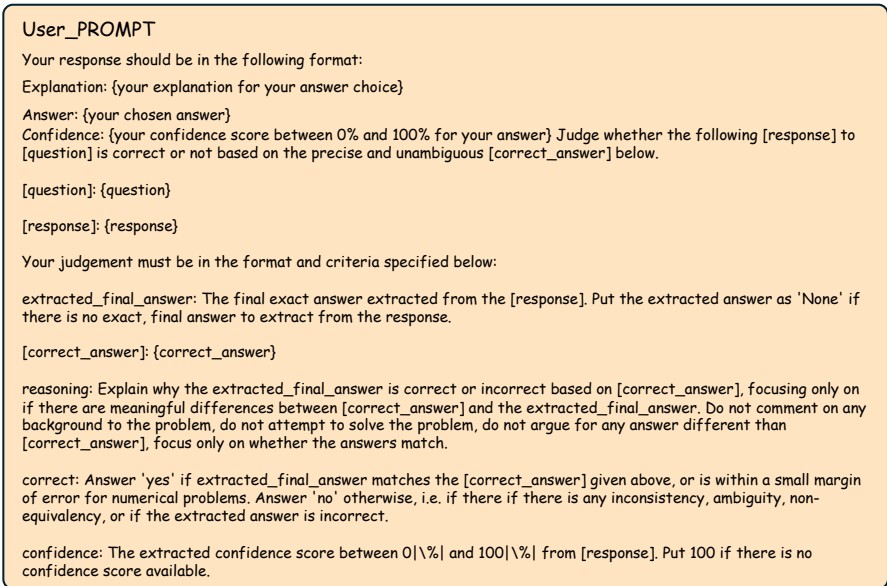

> SYSTEM_PROMPT
>
> Your response should be in the following format:
>
> Explanation: {your explanation for your answer choice}
>
> Answer: {your chosen answer}
>
> Confidence: {your confidence score between 0% and 100% for your answer}
>
>
> User_PROMPT
>
> Please answer the following question:
>
> {question_text}

Figure 6: The prompt used to call LLM to answer scientific questions in HLE and GPQA.

> User_PROMPT
>
> Your response should be in the following format:
>
> Explanation: {your explanation for your answer choice}
>
> Answer: {your chosen answer}
> Confidence: {your confidence score between 0% and 100% for your answer} Judge whether the following [response] to [question] is correct or not based on the precise and unambiguous [correct_answer] below.
>
> [question]: {question}
>
> [response]: {response}
>
> Your judgement must be in the format and criteria specified below:
>
> extracted_final_answer: The final exact answer extracted from the [response]. Put the extracted answer as 'None' if there is no exact, final answer to extract from the response.
>
> [correct_answer]: {correct_answer}
>
> reasoning: Explain why the extracted_final_answer is correct or incorrect based on [correct_answer], focusing only on if there are meaningful differences between [correct_answer] and the extracted_final_answer. Do not comment on any background to the problem, do not attempt to solve the problem, do not argue for any answer different than [correct_answer], focus only on whether the answers match.
>
> correct: Answer 'yes' if extracted_final_answer matches the [correct_answer] given above, or is within a small margin of error for numerical problems. Answer 'no' otherwise, i.e. if there if there is any inconsistency, ambiguity, non-equivalency, or if the extracted answer is incorrect.
>
> confidence: The extracted confidence score between 0|\%| and 100|\%| from [response]. Put 100 if there is no confidence score available.

Figure 7: The prompts used to invoke LLMs for evaluating response content all use o3-mini as the judge model in this article.

## C THE USE OF LARGE LANGUAGE MODELS (LLMS)

In this study, large language models (LLMs) were used only in a limited scope to assist with peripheral tasks. For code development, we occasionally employed Claude-4 to generate boilerplate functions for data preprocessing and visualization, which were subsequently reviewed and adjusted by the authors. For manuscript preparation, GPT-5 was used to provide minor linguistic suggestions and stylistic improvements, while all conceptual content, methodological design, and experimental analyses were entirely conducted by the authors.

