# OpenReview forum: "DualResearch: Entropy-Gated Dual-Graph Retrieval for Answer Reconstruction"
_ICLR.cc/2026/Conference — ICLR 2026 Conference Withdrawn Submission_

### Official Review · Reviewer_63yC · 2025-10-28

**Soundness:** 3
**Presentation:** 3
**Contribution:** 2
**Rating:** 4
**Confidence:** 3

**Summary:**

This paper presents DualResearch, a dual-graph retrieval and fusion framework that enhances reasoning systems integrating Large Language Models (LLMs) with external tools. The central idea is to jointly model two sources of evidence: a Breadth Semantic Graph (BG) that captures stable, cross-document background knowledge, and a Depth Causal Graph (DG) that records reliable, execution-based procedural traces from tool interactions. Each graph produces an answer distribution, and these are adaptively fused through an entropy-gated mechanism that calibrates confidence and balances semantic versus causal evidence. Experiments on the Humanity’s Last Exam (HLE) and GPQA-Diamond benchmarks demonstrate consistent improvements over the baseline InternAgent and outperform several recent deep-research systems.

**Strengths:**

- The separation into breadth (semantic) and depth (causal) graphs seems novel for tool-augmented LLMs.
- Experimental results on HLE and GPQA benchmarks demonstrate consistent improvements over baseline in different domains.
- The dual-graph design produces verifiable evidence chains, addressing transparency issues,  which are bottlenecks in current scientific and reasoning-oriented LLM systems.

**Weaknesses:**

- The experiments are restricted to the HLE and GPQA benchmarks. Broader evaluation on datasets such as BrowseComp or GAIA would strengthen the claim of generalizability across domains and tasks.
- The paper omits several recently released deep-research systems, such as TongYi-DeepResearch and Kimi-Researcher, that were available before submission. Including these would provide a more comprehensive and fair comparison of competitiveness.
- The proposed framework assumes access to well-structured, high-quality execution logs from agents. In realistic scenarios with noisy or incomplete logs, the system’s performance and stability may degrade, raising concerns about robustness in uncontrolled environments.
- Building and maintaining dual graphs with per-layer relevance and edge reliability may be computationally intensive for large-scale or streaming research workflows. The paper lacks runtime or scalability analysis to justify feasibility for real-world deployment.
- While the case studies emphasize successful instances, the paper provides little insight into failure cases, such as when semantic and causal graphs conflict or when entropy gating misjudges evidence balance.

**Questions:**

See weaknesses.

---

> ### Author Response · Authors · 2025-12-01
> **Respones to Benchmark and Sota**
>
> Thank you very much for your helpful suggestions.
>
> Following your advice, we have conducted experiments on a broader range of benchmarks to validate the effectiveness of our DualResearch framework. Due to time constraints, we first added results on the more commonly used GAIA benchmark:
>
> | Data Setting | Model                          | Level-1    | Level-2   | Level-3     | Avg.       |
> |--------------|--------------|-----------------|------------|-----------|-------------|
> | **GAIA**     | **o4-mini**  | InternAgent     | 78.49      | 69.18     | 26.53       | 65.12      |
> |              |              | DualResearch    | 84.95      | 73.58     | 36.73       | 71.10      |
> |              |              | **Improvement** | **↑6.54** | **↑4.40** | **↑10.20** | **↑5.98** |
>
> ---
>
> | Method                         | Level-1| Level-2| Level-3| Avg.   |
> |--------------------------------|--------|--------|--------|--------|
> | Qwen3-235B†                    | 15.09  |3.49    | 3.84   | 6.67   |
> | o4-mini †                      | 28.30  | 12.79  | 7.69   | 16.97  |
> | Deepseek-R1 †                  | 33.96  | 13.95  | 3.84   | 18.78  |
> | OpenAI Deep Research ‡         | 74.29  | 69.06  | 47.60  | 67.36  |
> | Tongyi ‡                       | -      | -      | -      | 70.90  |
> | InternAgent (o4-mini) ‡        | 78.49  | 69.18  | 26.53  | 65.12  |
> | **DualResearch (o4-mini) ‡**   | 84.95  | 73.58  | 36.73  | 71.10  |
>
> Taken together, the results on GAIA show that DualResearch can still systematically improve the performance of baseline agents under the same logs and backbone models across different datasets and task formats, with gains being more pronounced on difficult, multi-step reasoning instances. This is consistent with the trends we observed on HLE and GPQA, and further confirms that dual-graph retrieval combined with entropy-gated aggregation is not a special case tailored to a single benchmark, but rather a structured evidence reconstruction method with a certain degree of generality across domains and tasks.
>
> In addition, we have included comparisons with Tongyi-DeepResearch and Kimi-Researcher in the revised version. It is important to emphasize that DualResearch itself is not a full-fledged deep-research agent, but a post-processing scheme applied to the execution logs of existing systems. Therefore, although on HLE Text-Only our average accuracy is slightly lower than that of the end-to-end Tongyi-DeepResearch and X-Masters, we are able to consistently improve over single-round inference and the original InternAgent results built on the same base model, and we achieve first or second place in most subject areas. This suggests that DualResearch has good transferability and plug-and-play properties, making it more suitable as a general enhancement module on top of current and future deep-research systems, rather than as a separate, fully competitive end-to-end solution.

---

> ### Author Response · Authors · 2025-12-01
> **Response to other questions**
>
> **Regarding the robustness issue you raised:** In our setting, the quality of the graphs does indeed depend on the logs themselves. For this reason, we do not construct graphs directly from raw InternAgent logs. Instead, we apply a structuring and cleaning pipeline that denoises, normalizes, and aligns tool calls and intermediate artifacts before using them as graph inputs. In the experimental section we also explicitly state: “we collected log files …”. The cleaning process is quite straightforward: a script extracts the relevant information from the logs and stitches it together into a complete document. This script will be open-sourced together with the entire project.
>
> **Regarding the issue of runtime efficiency:** DualResearch is engineered so that dual-graph construction and scoring scale with the size of a single research trace rather than with the size of the entire corpus or tool universe. Concretely, both the Breadth Semantic Graph and the Depth Causal Graph are built from cleaned logs of one deep-research run. The number of nodes is linear in the number of retrieved snippets, intermediate artifacts, and tool calls, and edge reliabilities are computed locally from validator success, typing/unit checks, and a small number of retries, without any global optimization or message-passing over the whole graph. The per-layer relevance functions are likewise lightweight. On the breadth side we only perform one-hop neighborhood smoothing followed by a single cosine similarity per node, explicitly avoiding multi-step diffusion or GNN-style propagation. On the depth side, path relevance is computed by dynamic programming on short typed operation sequences and restricted to admissible paths of bounded length, so the dominant cost is linear in the number of candidate paths with a small constant factor.
>
> | Metrics     | Average of InternAgent | Maximum of DualResearch     | Minmun of DualResearch   | Average of DualResearch   |
> |-------------|------------------------|----------------------------:|-------------------------:|--------------------------:|
> | Time（min）  | 17.64                  | 6.16                        | 1.13                     | 2.81                      |
> | Token（K）   | 1413.51                | 96.43                       | 30.17                    | 53.74                     |
>
> Empirically, this design leads to a clear runtime advantage over simply running the agent again. On HLE Text-Only, the baseline InternAgent requires on average 17.64 minutes and 1413K tokens per problem, whereas DualResearch averages 2.81 minutes and 53.74K tokens; even for the heaviest instances, our time and token usage remain well below the baseline.
>
> In large-scale or streaming workflows, the same locality properties make incremental maintenance natural: new tool calls and evidence snippets are appended as nodes and edges with local updates to neighborhood embeddings and a small set of affected paths, without rebuilding the whole graph. For production deployment, the dual graphs can be hosted in an industrial graph database (e.g., Neo4j), where indexing and neighborhood queries are handled by the database engine and DualResearch only performs light-weight scoring on retrieved subgraphs. Together with the observed gains in both performance and efficiency on HLE and GPQA, these algorithmic and systems considerations support the feasibility of applying DualResearch in real-world, large-scale deep-research pipelines.
>
> **Regarding potential conflicts between the two graphs:** Conceptually, the Breadth Semantic Graph and the Depth Causal Graph are designed to be complementary rather than competing predictors. The breadth-side graph stores static background entities, paragraphs, and symbolic formula tokens, connected by lightweight semantic/evidential edges. The depth-side graph abstracts execution provenance into action–artifact–validator chains and employs a matching score that is sensitive to order and type. Therefore, there will be no direct structural conflict between the two.
>
> In practice, the main sources of divergence we observe do not stem from logical conflicts between semantics and causality, but from noise introduced by certain linguistic phenomena, such as rare terminology, polysemy, or ambiguous expressions. These factors may cause the breadth graph to over-activate spuriously related semantic neighborhoods.
>
> To mitigate this issue, we explicitly introduce an Offtopic penalty term on breadth-side paths to suppress semantic drift. We then map both channels into answer distributions and fuse them in log space via entropy-driven gating. This entropy gate down-weights channels that are high-entropy and internally inconsistent, and further applies a global calibration term to avoid overconfidence when both channels are uncertain. The results of the ablation experiments further demonstrate that under the aggregation effect, the two graphs can achieve a mutual interaction gain.

---

### Official Review · Reviewer_QuQf · 2025-10-30

**Soundness:** 2
**Presentation:** 2
**Contribution:** 2
**Rating:** 2
**Confidence:** 3

**Summary:**

This paper proposes DualResearch which is designed to enhance the reliability and evidence quality of answers generated by multi-step, tool-intensive deep-research systems. The method's core strength lies in jointly modeling two complementary structures: a breadth semantic graph encoding stable background knowledge and a depth causal graph tracking the execution provenance of the reasoning chain.. The authors propose an entropy-gated rule to fuse the distributions from two graphs, which dynamically up-weights the information channel exhibiting higher certainty, thus mitigating context pollution. The experiments  demonstrate stable and effective performance gains on scientific reasoning benchmarks like HLE and GPQA, suggesting DualResearch is a valuable post-processing complement to existing complex reasoning frameworks.

**Strengths:**

a. The DualSearch framework jointly models a breadth semantic graph and a depth causal graph, capturing both stable background knowledge and reasoning provenance.

b. The critical entropy-gated fusion rule dynamically enhances the reliability of the final answer by prioritizing the information channel (breadth or depth) that exhibits higher certainty, effectively mitigating context pollution and weak evidence.

c. The method achieves stable performance gains on challenging scientific reasoning benchmarks HLE and GPQA by reusing the log of the baselines.

**Weaknesses:**

a. DualResearch is explicitly presented as more of a "post-processing complement" to existing deep-research systems (like InternAgent). While this is valuable, it means the framework is not a standalone reasoning method. This makes it difficult to assess its direct performance gain versus a fundamentally better end-to-end LLM/tool-use setup.

b. The success of the proposed method largely depends on the quality of both the semantic graph  and the causal graph. This quality is influenced by the log file from the baseline and also the method of graph construction. How to ensure the quality of graphs during construction?

c. X-masters have provided their score on HLE in each subset in https://github.com/sjtu-sai-agents/X-Master. But the authors only reported the score on Bio/Med subset and claim achieving SOTA or second-best performance. X-Master has higher performance in MATH and chemistry compared with DualResearch.

**Questions:**

a. How to construct the breath semantic graph and the depth causal graph?

b. How to compute the confidence s_{b}(e) and s_{d}(e)?

c. Will the code be open-sourced for reproductivity？

---

> ### Author Response · Authors · 2025-12-01
> **Response to Weaknesses**
>
> Thank you very much for your valuable comments.
>
> **For Weakness a:** To the best of our knowledge, there is currently no dedicated work that performs post-processing for DeepResearch-style methods. Therefore, we introduce a generic ReAct-based procedure as a post-processing baseline. For this baseline, we feed in the same questions and log contents. We conduct a comparison on HLE Text-Only, and the results are as follows:
>
> | Method                              | Math        | Bio/Med        | CS/AI        | Physics       | Human.       | Chem.         | Engineer.        | Other        | Avg.        |
> | :-----------------------------------| ----------: | -------------: | -----------: | ------------: | ----------:  | ------------: | ---------------: | ----------:  | ----------: |
> | InternAgent (o4-mini)              | 23.5        | 18.9           | 13.9         | 17.3          | 21.6         | 21.6          | 18.8             | 25.0         | 21.3        |
> | ReAct (o4-mini)                    | 25.5        | 17.6           | 10.7         | 10.4          | 15.0         | 25.7          | 28.1             | 18.2         | 20.3        |
> | DualResearch (o4-mini)             | **31.3**    | **27.0**       | **26.8**     | **24.3**      | **29.0**     | **25.7**      | **28.1**         | **30.1**     | **29.0**    |
>
> It is clear that generic ReAct post-processing does not yield overall gains, whereas DualResearch achieves substantial improvements across all subject areas: on average, it improves accuracy by 7.7 percentage points over InternAgent and by 8.7 percentage points over ReAct. The advantage is particularly pronounced in domains such as CS/AI and Bio/Med, which demand complex reasoning and the integration of multi-source evidence. This indicates that dual-graph modeling combined with entropy-gated aggregation can more effectively filter reliable evidence and suppress noise from execution trajectories.
>
> **For Weakness b:** In our setting, the quality of the graph does indeed depend on the logs themselves. For this reason, we do not construct graphs directly from the raw InternAgent logs. Instead, we introduce a structuring and cleaning pipeline to denoise, normalize, and align tool calls and intermediate artifacts before using them as graph inputs. In the experimental section we explicitly state that “we collected log files …”. The cleaning process itself is very straightforward: a script extracts the relevant information from the logs and concatenates it into a complete document. This script will be open-sourced together with the overall project.
>
> **For Weakness c:** At the time of our original submission, X-Master had not yet released more of its own results. Following your suggestion, we have added the corresponding results in Table 2 of the revision. We observe that X-Masters performs better than DualResearch overall on Mathematics, Bio/Med, and Human, whereas DualResearch achieves stronger performance on CS/AI, Physics, Engineer, and Other subsets, revealing complementary strengths on different types of tasks. In terms of overall average score, X-Masters reaches 32.1, higher than the 29.0 of DualResearch. However, the former relies on multi-channel ensembling and multiple model invocations for majority voting, while the latter only performs graph construction and aggregation over existing research logs, without increasing the number of reasoning rounds.

---

> ### Author Response · Authors · 2025-12-01
> **Response to Questions**
>
> Thank you for your valuable opinions.
>
> **For Question a:** The framework builds two complementary graphs that encode distinct aspects of evidence. The Breadth Semantic Graph $G_B=(V_B,E_B,s_B)$ contains canonical entities, paragraph or table spans, and formula symbols as nodes, with lightweight semantic and evidential relations as edges; each edge e is assigned a normalized confidence $s_B(e)$ summarizing extraction reliability and cross-source support. In parallel, the Depth Causal Graph $G_D=(V_D,E_D,s_D)$ abstracts execution provenance into typed nodes for actions, artifacts or results, and validators; directed edges record consumes, produces, verified-by, and carryover dependencies and are admitted only when typing, units, and temporal order are coherent, after which they receive a single confidence $s_D(e)$ derived from validator success and repeatability. The two substrates are queried with layer-native encoders: the query encoder $f(\cdot)$, a breadth encoder $g_B(\cdot)$ with one-hop confidence-weighted smoothing $\bar g_B(\cdot)$, and a depth encoder $g_D(\cdot)$ whose signals are used together with type and order constraints to respect procedural structure. This separation preserves stable background anchors while capturing short, auditable chains of operations in logs.
>
> **For Question b:** To address the issue of unclear description at this point, we have revised the wording in the updated manuscript. In the Breadth Semantic Graph, an edge $e \in E_B$ corresponds to a lightweight semantic or evidential relation over static background knowledge, such as mentions, defines, cites, etc. In the main text, we uniformly denote its score by $s_B(e) \in (0,1]$, which summarizes both the reliability of the extraction and the degree of consistent support across sources. In implementation, each candidate edge is first produced by one or multiple runs of relation extraction and entity linking. We record the extraction model’s score for this relation and the number of times it is independently re-extracted from different log segments. We then linearly or sub-linearly combine these signals into a raw score and apply a monotonic normalization within the task scope so that all breadth-edge scores fall into the open interval $(0,1]$, yielding $s_B(e)$. Since the scores of breadth-side paths are accumulated in logarithmic form in Eq. (3), this normalization only needs to be monotonic, ensuring that edges that are more reliable and more consistently supported across sources receive larger contributions at the path level, while the specific numeric scale does not alter the relative ranking of paths.
>
> In the Depth Causal Graph, an edge $e \in E_D$ abstracts a causal dependency in the execution trace, for example that a result is produced by a particular action, and it is required to satisfy consistency in type, unit, and temporal order. In our definition, we retain only edges that pass these hard constraints and assign to each a single confidence score $s_D(e) \in (0,1]$, derived from the validator’s success behavior and its reproducibility under retries and branching. Specifically, for each candidate dependency, we count its occurrences across different execution branches and retry attempts, as well as the pass rate of the corresponding validator, and again apply a monotonic normalization to map it into $(0,1]$, which we use as $s_D(e)$.
>
>
> **For Question c:** We will definitely release our code in the future. We also hope that it can be helpful for subsequent research.

---

### Official Review · Reviewer_8gvY · 2025-10-31

**Soundness:** 3
**Presentation:** 1
**Contribution:** 3
**Rating:** 4
**Confidence:** 4

**Summary:**

This paper introduces DualResearch, a framework designed to enhance tool-intensive scientific reasoning systems. The core innovation is the joint modeling of two complementary graphs: a Breadth Semantic Graph for stable background knowledge and a Depth Causal Graph for procedural provenance from tool execution logs. The framework employs an entropy-gated fusion mechanism to reconcile the heterogeneous evidence from these graphs, adaptively weighting the more certain channel.

**Strengths:**

1）The proposed dual-graph architecture is a novel approach to addressing the distinct challenges of semantic coverage and causal consistency in complex reasoning. The layer-native retrieval functions (e.g., seed-anchored semantic diffusion for breadth, causal-semantic path matching for depth) are well-motivated and tailored to their respective graph structures.

2）The entropy-gated fusion mechanism provides a theoretically grounded method for combining evidence from heterogeneous sources. The empirical results, showing consistent and significant improvements over strong baselines by reusing their execution logs, are compelling and highlight the practical utility of the method.

**Weaknesses:**

1）While the paper heavily emphasizes reproducibility and verifiability as key advantages, it does not sufficiently clarify the specific application scenarios where these properties are critically demanded. In many LLM applications, diversity and creativity are valued over strict reproducibility. The argument for this as a primary innovation would be stronger if supported by clear examples of domains (e.g., clinical decision support, regulatory compliance) where reproducible reasoning chains are non-negotiable.

2）The readability of the paper is hampered in several sections. The descriptions of the graph construction, path scoring, and fusion mechanics are highly technical and dense, making them difficult to follow without more illustrative examples and intuitive explanations.

**Questions:**

1）The paper argues that "retrieval and aggregation should reflect the epistemological structure of the task" as a direct response to the limitations of deep-research systems. However, the logical leap from the mentioned issues (e.g., noisy retrieval, missing causal constraints) to this specific conclusion feels abrupt. If the cited work by Huang et al. (2025) explicitly identifies the lack of a step-by-step logic chain as a root cause for hallucinations or errors, this connection should be made explicit.

2）The depth similarity in Section 3.1 is conceptually sound but abstract. Providing a concrete, minimal example in the main text (or a detailed one in the appendix) comparing the operation sequences for a query and a candidate path would greatly enhance comprehension.

3）Line 181 lists six relation types for the Breadth Graph. What are the relative proportions of these relation types in the constructed graphs? Is the coverage of these six types comprehensive, or is there a long tail of other relations that were excluded?

4）There appears to be a typo in Table 1, where two rows are labeled "HLE Text-Only." This should be corrected for clarity; presumably one is meant to be "HLE All-Set."

---

> ### Author Response · Authors · 2025-12-01
> **Response to Weaknesses**
>
> Thank you for your comments. We fully agree with your view that in certain settings, creativity is the more desirable trait. However, in this paper we aim to emphasize that, in tool-intensive tasks, especially scientific reasoning workflows involving external retrieval and code execution, a stable reasoning process and a replayable evidence chain are often non-negotiable quality requirements.
>
> As illustrated in Figure 1 with the “Turing machine halting steps” example, a traditional deep-research pipeline arrives at an incorrect conclusion due to noisy retrieval and the absence of causal constraints. In contrast, DualResearch, via its structured process graph and entropy-gated aggregation, corrects the conclusion while also making explicit a verifiable “analyze–simulate–compare” chain. Although such tasks are not themselves in healthcare or regulatory domains, they embody the same class of requirements: answers must not only be correct, but also amenable to review and replay.
>
> ---
>
> In order to better elaborate on the research methods of this paper, we have improved the Method in the submitted Revision.
>
> **Graph construction**
> The framework builds two complementary graphs that encode distinct aspects of evidence. The Breadth Semantic Graph $G_B=(V_B,E_B,s_B)$ contains canonical entities, paragraph or table spans, and formula symbols as nodes, with lightweight semantic and evidential relations as edges; each edge e is assigned a normalized confidence $s_B(e)$ summarizing extraction reliability and cross-source support. In parallel, the Depth Causal Graph $G_D=(V_D,E_D,s_D)$ abstracts execution provenance into typed nodes for actions, artifacts or results, and validators; directed edges record consumes, produces, verified-by, and carryover dependencies and are admitted only when typing, units, and temporal order are coherent, after which they receive a single confidence $s_D(e)$ derived from validator success and repeatability. The two substrates are queried with layer-native encoders: the query encoder $f(\cdot)$, a breadth encoder $g_B(\cdot)$ with one-hop confidence-weighted smoothing $\bar g_B(\cdot)$, and a depth encoder $g_D(\cdot)$ whose signals are used together with type and order constraints to respect procedural structure. This separation preserves stable background anchors while capturing short, auditable chains of operations in logs.
>
> **Path scoring**
> Retrieval and path relevance are evaluated in a graph-native manner. On the breadth side, a node $v$ is scored by a cosine between the query embedding and the one-hop neighborhood–smoothed representation of $v$, $S_B(v\mid q)=\cos!\big(f(q),\bar g_B(v)\big)$, which rewards terms supported by nearby evidence and suppresses matches to isolated nodes. On the depth side, order and typing are enforced by comparing the sequence of required operations extracted from the query, $O(q)$, to the sequence on a short admissible path p that passes unit, type, and time gates; the target score maximizes over paths ending at $t$,
>
> $S_D(t\mid q)=\max_{p\in\mathcal P_{\le L}(t)} R(p)\cdot \frac{\mathrm{LCS}^\dagger(O(q),O(p))}{|O(q)|},$
>
> with path reliability $R(p)=\big(\prod_{e\in p} s_D(e)\big)^\tau$. For path-level scoring with drift control, a breadth path p receives a log-additive score that trades off semantic support and topical focus,
>
> $S_B(p\mid q)=\sum_{e\in p}\log w_B^e-\lambda_{\text{off}}\cdot \mathrm{Offtopic}(p),$
>
> and an analogous score $S_D(p\mid q)$ incorporates edge direction, type, and temporal consistency on $G_D$. These criteria prefer brief, reliable chains that match the procedural intent of the query while penalizing semantic wandering in breadth exploration.
>
> **Fusion mechanics**
> Evidence from both graphs is converted into per-channel answer distributions by aggregating over the supporting path sets $P_B(a)$ and $P_D(a)$ using log-sum-exp, yielding $P_B(a\mid q)$ and $P_D(a\mid q)$. Channel uncertainty is quantified by Shannon entropies $H_B$ and $H_D$. The final posterior uses an entropy-gated geometric mixture in log space,
>
> $P(a\mid q)=\mathrm{softmax}\big(\alpha(H)\log P_D(a\mid q)+\big(1-\alpha(H)\big)\log P_B(a\mid q)\big),
>  \quad \alpha(H)=\frac{e^{-H_D}}{e^{-H_D}+e^{-H_B}},$
>
> followed by a global calibration step with temperature $\gamma$ and penalty $\beta$ to discourage overconfidence and a MAP selection $a^\star=\arg\max_a \tilde P(a\mid q)$. This rule up-weights the sharper channel, attenuates diffuse evidence, and amplifies agreement when both graphs are confident; it also enables extraction of a minimal verifiable chain by greedily retaining edges with the largest marginal contribution to $\tilde P(a^\star\mid q)$. Theoretical analysis further shows that such entropy-gated log-linear fusion upper-bounds pointwise loss by a convex combination of channel losses and can generalize at least as well as the better single channel under entropy–loss calibration, which explains the robustness observed in experiments.

---

> ### Author Response · Authors · 2025-12-01
> **Response to Questions**
>
> Thank you for your questions; they are highly valuable to us.
>
> **Regarding Question 1**, following your suggestion we have refined the introduction in the revised manuscript. Our intention is to argue that these observable failures in fact share a common underlying mechanism: the system treats all evidence as a homogeneous pool of text, while ignoring the fact that scientific reasoning relies on epistemic resources that are heterogeneous in nature. In the introduction, we first show that even when relevant content has been retrieved, a deep research agent may still arrive at incorrect conclusions due to context pollution and the lack of explicit intermediate reasoning steps. These errors stem precisely from the fact that retrieval is defined purely by semantic similarity, whereas aggregation is performed on lengthy and unstructured logs. This means that background facts, tool calls, and other elements are all mixed into a single textual channel.
>
> We propose that “retrieval and aggregation should reflect the epistemic structure of the task” because, at the retrieval stage, the system must distinguish between stable declarative knowledge and transient procedural trajectories; at the aggregation stage, it must reason over these two types of evidence under appropriate constraints, rather than simply flattening them into an undifferentiated bag of paragraphs. To concretize this principle, we introduce a Breadth Semantic Graph to represent static background knowledge and a Depth Causal Graph to represent process-level provenance, and then integrate information through posterior-level fusion rather than by naively concatenating the raw text.
>
> We cite the work of Huang et al. (2025) to align our motivation with their analysis of deep research agents. They systematically document that such systems often lack explicit and verifiable step-by-step reasoning chains, and point out that when long tool-use trajectories are treated as undifferentiated context rather than as a structured, replayable process, the system becomes prone to hallucinatory outputs and brittle behavior.
>
> **Regarding Question 2**, we have refined the Method section and provided an illustrative example in the appendix to improve the description of depth similarity, with the aim of presenting our proposed method more clearly.
>
> **Regarding Question 3**, when constructing the Breadth Graph, all relations between entities are constrained to the following six types: mentions, defines, aliases, cites, supports, and derived from. In other words, these six relations cover the entire graph. Specifically, we compute the proportions of these relations on the Human and Engineer subsets of the HLE dataset as follows：
>
> | Category | mentions | defines | aliases | cites | supports | derived from |
> | :--- | ---: | ---: | ---: | ---: | ---: | ---: |
> | Human. | 33.51 | 14.64 | 3.13 | 22.14 | 12.42 | 14.16 |
> | Engineer. | 37.18 | 19.31 | 1.71 | 17.42 | 9.83 | 14.55 |
>
> **Regarding Question 4**, we have corrected the erroneous labels in Table 1; you may refer to the submitted revision for the detailed changes.

---

### Official Review · Reviewer_j5EC · 2025-11-01

**Soundness:** 3
**Presentation:** 1
**Contribution:** 3
**Rating:** 4
**Confidence:** 3

**Summary:**

This work builds on compound systems (iterative / multi-agent) for solving complex tasks that require tool calls for pulling external information and performing complex data manipulation. In particular, it attempts to improve systems on Humanity's Last Exam (HLE) and Google-Proof Q&A (GPQA).

The paper proposes an approach to review the logs of existing compound research systems (InternAgent, X-Masters) and refine them using a two-channel scoring mechanism. In particular:
1. DualResearch takes the execution trace of an existing research system (InternAgent in the main paper) and extracts two graphs with edges scored by confidence of extraction:
    1. Breadth Semantic Graph: Nodes are entities, paraphras, table spans. Edges show how the spans relate to each other
    2. Depth Causal Graph: Graph nodes are tool calls, results and validator outputs. Edges link tool calls to corresponding outputs and so on.
2. On the extracted graph, the method first retrieves a subgraph based on relevance to the question. The relevance function on the breadth graph uses node similarity to the query. The relevance function on the depth graph measures the alignment of paths to the query.
3. Each candidate answer entity is scored by using path scores from both graphs and computing a weighted average using the uncertainty in each channel. (Theoretical results show that the uncertainty weighted average achieves a lower regret than using each channel individually)
4. The scoring process is used to extract supporting evidence in terms of the highest scored path that leads to the answer entity.

The method is applied to research logs from InternAgent and X-Masters, and shows consistent improvements with different backbone LLMs. A case study is provided to show how DualResearch compresses the logs of InternAgent for further processing. Ablations show the gains from combining the two channels (each channel on its own is not able to improve significantly over the baseline).

The biggest weakness of the paper is that a lot of the details are vaguely described. Moreover, the total additional LLM calls are not discussed.

**Strengths:**

1. DualResearch operates on a compressed and structured representation of the execution trace of existing agents and uses this representation to improve accuracy
    - The case study highlights how the log space of the original agent can be compressed significantly (10x fewer tokens) to reduce drift and erroneous reasoning
2. Ablations show that the full scoring mechanism (uncertainty weighted average of scores from both channels) is necessary to improve over the baseline agent
3. Additional Experiments that show the effect of building subject level graphs are interesting

**Weaknesses:**

1. Several components in the paper are not described in detail. (See questions below).
    - Most importantly, the graph extraction module is not described. Is it general or does it need to be specifically designed for each base agent?
    - I believe that the proposed structured representation is a major contribution of the paper but it is not described with enough detail to be reproducible.
2. The paper assumes that supporting evidence exists as a chain of steps. But if there are multiple query entities, won't the evidence for a DAG (query entities to the answer entity) instead of a chain? It is unclear how evidence paths are defined.
3. What is the run-time of the DualResearch system? How many additional LLM calls are required per query?
    - DualResearch works on top of execution traces of other systems. However, none of the baselines can refine the logs. I believe DualResearch will perform better than naive methods that try to refine the logs but this is not established or discussed.

**Questions:**

Clarifications about the method
---
1. Is it correct to say that the method cannot improve over the baseline agent if the correct answer is not an entity in the execution log?

Clarifications related to Weakness 1
---
2. Sec 3.1: How are the two graphs extracted from the logs?
3. Sec 3.1: What are the encoding functions for the query and the graph nodes?
4. Line 206: How is $O(q)$ extracted from the query?
5. Eq 2: What is $t$ and $\mathcal{P}_{\leq L}(t)$?
6. Sec 3.2: This step assumes that candidate answer entities are known. But not all queries in HLE have choices provided. How do you define the answer set in such cases?
7. Sec 3.2: How do you define the set of paths $P_B(a)$ and $P_D(a)$? Are all possible non-asnwer entities considered as starting points?
8. Eq 3: What is $w_e^B$?
9. Eq 3: How is the Offtopic score calculated?
10. Eq 4: How many terms are in the sum over paths?
11. Line 408: How would DualResearch without entropy-based aggregation enlarge the solution space? The ablation should have no effect on the set of candidate answers.

Clarifications related to Weakness 3
---
12. How many additional LLM calls are required in DualResearch starting from the InternAgent log?
13. Can you provide a discussion on how DualResearch is better than naively refining the logs with additional validation agent calls?

Other clarifications
---
14. Line 198: What is lightly averaged?
15. Line 203: What does it mean by "This one-hop smoothing avoids multi-step diffusion and hyperparameters". i.e. what does avoid hyperparameters mean?
16. Line 218: Are you referring to multi-step diffusion in the original execution log?
17. What is Signal graph? I am not familiar with this terminology. It seems to be related to building individual graphs per query.

---

> ### Author Response · Authors · 2025-12-01
> **Response to Weakness 1**
>
> Thank you for your valuable comments. Below we provide a response to Weakness 1:
>
> **The Graph Extraction Module is a general paradigm that converts raw log records into structured representations. Its detailed operation is as follows (we have also revised Sec. 3 in the main text accordingly):**
>
> **Breadth semantic graph: entity–relation extraction.**
> For each segment of background knowledge retrieved from the logs, we feed it into a prompt template for an LLM, asking the model to extract all important entities (concepts, numerical values, units, etc.) and to identify the semantic relations among them. The outputs, which are similar to triples, form the nodes $V_B$ and edges $E_B$ of the breadth semantic graph. The extraction process uses a fixed prompt template to ensure systematic coverage. Each extracted edge is assigned a confidence score $s_B(e)$ (see Definition 1), which reflects the reliability of the extraction.
>
> **Depth causal graph: behavioral trajectory extraction.**
> For each record in the logs that contains an agent action, tool call, or verification step, we use an LLM to parse it into the following structured elements: “Given this log entry, please list: (i) the action/tool and its parameters; (ii) the input artifacts; (iii) the output artifacts; and (iv) what verifications were performed.” The parsed result yields one Action node and several Artifact nodes, which are connected via consumes/produces edges. If the log contains consistency checks or unit checks, we create a Validator node and a verified-by edge. We only accept an edge when type/unit/time constraints are satisfied, thereby ensuring the coherence of the graph.
>
> **Encoding functions.**
> In Sec. 3.1, we explicitly define three types of encoding functions. Query encoder $f(\cdot)$, which encodes the query $q$ into a vector. Breadth-graph node/edge encoder $g_B(\cdot)$. During scoring, we use its one-hop neighborhood–smoothed representation $\bar g_B(v)$, which performs a light confidence-weighted averaging of neighboring node embeddings based on edge confidences. Depth-graph node/edge encoder $g_D(\cdot)$. Depth relevance is then combined with type/order-consistent path matching.
> In our implementation, we use **bge-small-en-v1.5**.
>
> **For short-answer questions in HLE**, we typically take three independent responses from a vanilla LLM as our candidate answer set.
>
> **Regarding the solution-space issue introduced by aggregation.**
> We agree that the ablation does not change the set of multiple-choice options, the candidate answer set (A) remains fixed. In Line 408, the phrase “enlarge the solution space” was imprecise. What we intended to convey is that, without entropy-based aggregation, more low-quality and mutually conflicting reasoning paths simultaneously support different options. This makes the posterior over (A) more diffuse and increases the model’s uncertainty, rather than literally introducing new candidates. We will revise the text to clarify this and replace “enlarge the solution space” with a more accurate description such as “increase the uncertainty and diffuseness of the posterior over the fixed candidate set.”
>
> **Notation and formulas:**
> * $O(q)$ denotes the set of action keywords obtained by parsing the question text with an LLM.
> * $t$ is the “target” node to be scored in the depth causal graph (typically the Artifact node corresponding to a candidate result/answer), and $\mathcal{P}_{\le L}(t)$ denotes the set of all “acceptable” directed paths ending at $t$ whose length does not exceed $L$.
> * The two path sets $P_B(a)$ and $P_D(a)$ are both constrained path collections that are seed-anchored to the query and end at candidate answer $a$. They start only from the seed node set that is strongly related to the question and options, performing constrained expansion from there, rather than freely wandering from arbitrary nodes in the graph.
> * $w_B^e$ is the weight of edge $e$ in the breadth graph, i.e., $w_B^e = s_B(e)$.
> * The Offtopic score is a *topic-drift penalty* used to downweight paths that gradually deviate from the query’s topic. In implementation, $\mathrm{Offtopic}(p)$ accumulates drift along the path. Intuitively, it sums the degree of off-topicness at each step, weighted by path length and edge confidence, so that long detours or cross-semantic-chain jumps receive stronger penalties. The $\log w_B^e$ term continues to reward semantic edges with strong supporting evidence; together they create a trade-off that suppresses off-topic excursions without penalizing necessary short hops.
> * The number of terms is exactly $|P_B(a)|$ or $|P_D(a)|$; it is not a fixed constant.
> * $\mathrm{Offtopic}(p)$ is the off-topic penalty of path $p$, used to reduce the impact of semantically drifting paths. It satisfies $\mathrm{Offtopic}(p) \ge 0$ and $\lambda_{\text{off}} \ge 0$.

---

> ### Author Response · Authors · 2025-12-01
> **Response to Weakness 2 & 3**
>
> Thank you for your valuable comments. Below I respond to Weakness 2:
>
> We do not assume that the evidence must a priori form a single linear chain. In fact, our depth graph is a directed acyclic graph (DAG), which can both branch and merge.
>
> When retrieving evidence, we search for acceptable paths within this DAG. After aggregation, we extract the minimal evidence chain for explanation by pruning the edges with the smallest contribution. In this way, the underlying evidence can be a general DAG, while what we present is a concise linear sub-path. We will clarify in the paper that DualResearch is capable of handling branched evidence graphs, rather than requiring the underlying domain itself to be linear.
>
> ---
>
> Thank you for your valuable comments. Below I respond to Weakness 3:
>
> **Regarding the efficiency of DualResearch:** in DualResearch, the additional LLM calls occur mainly in the stages of structurally extracting the execution logs and constructing the dual graphs. Their cost scales with the length of the input logs. Therefore, to assess the efficiency of our method, we focus on the total number of tokens consumed and the actual running time. To this end, we have measured and reported the running time and token usage of DualResearch on HLE Text-Only.
>
> | Metrics     | Average of InternAgent | Maximum of DualResearch     | Minmun of DualResearch   | Average of DualResearch   |
> |-------------|------------------------|----------------------------:|-------------------------:|--------------------------:|
> | Time（min）  | 17.64                  | 6.16                        | 1.13                     | 2.81                      |
> | Token（K）   | 1413.51                | 96.43                       | 30.17                    | 53.74                     |
>
> As shown in the table, compared with directly re-running InternAgent, DualResearch reduces the average inference time on HLE Text-Only from 17.64 minutes to 2.81 minutes, and the average token consumption from 1413K to 53.74K. Even in the worst case, the time and token cost of DualResearch are still clearly lower than those of the baseline. This indicates that, while maintaining or even improving accuracy, dual-graph reconstruction and log-based answer regeneration provide a more lightweight and more scalable way to reuse deep research, rather than simply “running the agent again.”
>
> **Comparison with a simple verifier agent:** your observation is very accurate. To the best of our knowledge, there is currently no dedicated work on post-processing for DeepResearch-style methods. Therefore, we introduce a generic ReAct-based procedure as a post-processing baseline. For this baseline, we feed in the same questions and execution logs. We conduct the comparison on HLE Text-Only, and the results are as follows:
>
> | Method                              | Math        | Bio/Med        | CS/AI        | Physics       | Human.       | Chem.         | Engineer.        | Other        | Avg.        |
> | :-----------------------------------| ----------: | -------------: | -----------: | ------------: | ----------:  | ------------: | ---------------: | ----------:  | ----------: |
> | InternAgent (o4-mini)              | 23.5        | 18.9           | 13.9         | 17.3          | 21.6         | 21.6          | 18.8             | 25.0         | 21.3        |
> | ReAct (o4-mini)                    | 25.5        | 17.6           | 10.7         | 10.4          | 15.0         | 25.7          | 28.1             | 18.2         | 20.3        |
> | DualResearch (o4-mini)             | **31.3**    | **27.0**       | **26.8**     | **24.3**      | **29.0**     | **25.7**      | **28.1**         | **30.1**     | **29.0**    |
>
> It is clear that generic ReAct post-processing does not yield overall gains, whereas DualResearch achieves substantial improvements across all subject areas: on average, it improves accuracy by 7.7 percentage points over InternAgent and by 8.7 percentage points over ReAct. The advantage is particularly pronounced in domains such as CS/AI and Bio/Med, which demand complex reasoning and the integration of multi-source evidence. This indicates that dual-graph modeling combined with entropy-gated aggregation can more effectively filter reliable evidence and suppress noise from execution trajectories.

---

> ### Author Response · Authors · 2025-12-01
> **Response to Other clarifications**
>
> Thank you for your valuable opinions. Below, I will respond to the Other questions you raised:
>
> **Lightly averaged** refers to how we construct the neighborhood-smoothed embedding $\bar g_B(v)$ in Eq. (1). We mix the node’s own embedding with a shallow, one-hop average over its immediate neighbors, weighted by edge confidences.
>
> **Avoid hyperparameters:** In many graph-based retrieval systems, node representations are refined through multi-step diffusion, repeatedly propagating signals across neighbors with tunable coefficients (such as propagation depth). These steps introduce extra hyperparameters  that require tuning and can make retrieval behavior unstable across datasets. In contrast, our one-hop smoothing averages each node’s embedding only with its immediate neighbors, weighted by their edge confidences. This achieves topology awareness but does not require specifying any of those diffusion hyperparameters. There’s no multi-hop expansion depth, no diffusion coefficient, and no iteration count. Hence, “avoids multi-step diffusion and hyperparameters” means:
> * we use a single, deterministic, local smoothing step;
> * we eliminate hyperparameter tuning associated with multi-hop or iterative propagation schemes.
>
> This design keeps the Breadth retrieval simple, robust to noise, and reproducible across domains, which was an intentional trade-off between semantic context enrichment and model interpretability.
>
> **Multi-step Diffusion** here does not refer to multi-step propagation on the original execution logs, but rather to multi-hop semantic diffusion on the Breadth Semantic Graph. As we explicitly state in Sec. 3.1, on the Breadth side we apply a “one-hop, neighborhood-smoothed” cosine similarity over nodes to obtain a topology-aware yet non-diffusive stable anchor. In contrast, on the Depth side we adopt short-path matching with order–type constraints and reliability weighting, instead of diffusion.
>
> **Signal graph** is our shorthand for the per-query (single-sample) graph we build from just that query’s solving log, without merging anything from other problems. In other words, every question gets its own self-contained graph; nothing is shared across questions. In the paper we contrast this with the Subject graph, which aggregates multiple problems from the same discipline into one larger, reusable graph.

---

### Note · Authors · 2026-01-06

I have read and agree with the venue's withdrawal policy on behalf of myself and my co-authors.